# Association among Vitamin D, Retinoic Acid-Related Orphan Receptors, and Vitamin D Hydroxyderivatives in Ovarian Cancer

**DOI:** 10.3390/nu12113541

**Published:** 2020-11-19

**Authors:** Anna A. Brożyna, Tae-Kang Kim, Marzena Zabłocka, Wojciech Jóźwicki, Junming Yue, Robert C. Tuckey, Anton M. Jetten, Andrzej T. Slominski

**Affiliations:** 1Department of Human Biology, Institute of Biology, Faculty of Biological and Veterinary Sciences, Nicolaus Copernicus University, 87-100 Toruń, Poland; 2Department of Dermatology, University of Alabama at Birmingham, Birmingham, AL 35294, USA; tkim@uabmc.edu; 3Department of Tumor Pathology and Pathomorphology, Oncology Centre—Prof. Franciszek Łukaszczyk Memorial Hospital in Bydgoszcz, 85-796 Bydgoszcz, Poland; marzenazablocka_73@wp.pl; 4Department of Tumor Pathology and Pathomorphology, Department of Oncology, Faculty of Health Sciences, Ludwik Rydygier Collegium Medicum in Bydgoszcz, Nicolaus Copernicus University, 87-100 Torun, Poland; jozwickiw@cm.umk.pl; 5Department of Pathology, University of Tennessee Health Science Center, Memphis, TN 38163, USA; jyue@uthsc.edu; 6School of Molecular Sciences, The University of Western Australia, Perth, WA 6009, Australia; robert.tuckey@uwa.edu.au; 7Cell Biology Section, Immunity, Inflammation and Disease Laboratory, National Institute of Environmental Health Sciences, National Institutes of Health, Research Triangle Park, NC 27709, USA; jetten@niehs.nih.gov; 8Laboratory Service of the VA Medical Center, Birmingham, AL 35294, USA

**Keywords:** ovarian cancers, vitamin D, nuclear receptors, VDR, RORα, RORγ

## Abstract

Vitamin D and its derivatives, acting via the vitamin D receptor (VDR) and retinoic acid-related orphan receptors γ and α (RORγ and RORα), show anticancer properties. Since pathological conditions are characterized by disturbances in the expression of these receptors, in this study, we investigated their expression in ovarian cancers (OCs), as well as explored the phenotypic effects of vitamin D hydroxyderivatives and RORγ/α agonists on OC cells. The VDR and RORγ showed both a nuclear and a cytoplasmic location, and their expression levels were found to be reduced in the primary and metastatic OCs in comparison to normal ovarian epithelium, as well as correlated to the tumor grade. This reduction in VDR and RORγ expression correlated with a shorter overall disease-free survival. VDR, RORγ, and RORα were also detected in SKOV-3 and OVCAR-3 cell lines with increased expression in the latter line. 20-Hydroxy-lumisterol3 (20(OH)L_3_) and synthetic RORα/RORγ agonist SR1078 inhibited proliferation only in the OVCAR-3 line, while 20-hydroxyvitamin-D_3_ (20(OH)D_3_) only inhibited SKOV-3 cell proliferation. 1,25(OH)_2_D_3_, 20(OH)L_3_, and SR1078, but not 20(OH)D_3_, inhibited spheroid formation in SKOV-3 cells. In summary, decreases in VDR, RORγ, and RORα expression correlated with an unfavorable outcome for OC, and compounds targeting these receptors had a context-dependent anti-tumor activity in vitro. We conclude that VDR and RORγ expression can be used in the diagnosis and prognosis of OC and suggest their ligands as potential candidates for OC therapy.

## 1. Introduction

Ovarian cancer (OC) is the seventh most frequent cancer in women and the leading cause of death in patients with gynecological cancer, with a higher incidence rate and poorer prognosis in less developed counties [1,2]. Since early OC is mostly asymptomatic, most cases (75%) are diagnosed at advanced stages [3]. The 5 year relative survival rate for invasive epithelial ovarian cancer patients is about 45% for all stages [1,2]. OC risk factors include older age, obesity, late pregnancy or nulliparity, in vitro fertilization, hormone replacement therapy based on estrogen alone, and genetic predisposition (for example, mutations in *BRCA1*, *BRCA2*, *PTEN*, *LH1*, *MLH3*, *MSH2*, *MSH6*, *TGFBR2*, *PMS1*, *PMS2*, *STK11*, *MUTYH*, and other genes [1]). At an early stage, OC is sensitive to current treatment regimens (5 year relative survival rate for stage I is 90%). Despite the improvement in treatment, OC patients develop recurrent disease very often, with an overall recurrence rate of 75%, up to 10–30% at early stages [3,4,5]. Although the 5 year relative survival rate has slightly improved, especially for Caucasians, high recurrence and mortality rates indicate the clear need for developing new methods for early diagnosis, recurrence prediction, and an efficient adjuvant treatment of OC.

It is widely appreciated that active forms of vitamin D can act as anticancer agents, inhibiting proliferation, stimulating differentiation and apoptosis, affecting tumor microenvironment, and modulating the immune response and sensitivity to antitumor therapies [6,7,8,9,10,11,12,13,14,15,16,17,18,19,20,21,22]. Vitamin D is formed in the skin after absorption of the ultraviolet B (UVB) energy by 7-dehydrocholesterol or is acquired via a food supplement [23,24,25,26,27,28,29,30,31]. In the canonical pathway, it is activated by two separate hydroxylation steps. The first step takes place in the liver and is catalyzed by 25-hydroxylases, cytochrome P450 (CYP)27A1 and CYP2R1, followed by the second hydroxylation at C1α in the kidneys catalyzed by CYP27B1, generating the hormonally active 1,25(OH)_2_D_3_ [17,32,33]. In recently discovered noncanonical pathways of activation, vitamin D3 or lumisterol is hydroxylated at C20 to produce 20-hydroxyvitamin D3 (20(OH)D3) or 20-hydroxylumisterol (20(OH)L_3_) as biologically active intermediates [34,35,36,37].

Epidemiological studies indicate that low vitamin D serum levels are related to higher OC risk and poorer prognosis [38,39]. Furthermore, decreased CYP27B1 expression in cancer cells is related to a poorer clinical outcome [40]. Active forms of vitamin D express anticancer properties via interaction with the vitamin D receptor (VDR) which is present in a variety of normal and cancerous tissues [17,18,41]. After binding 1,25(OH)_2_D_3_, the VDR heterodimerizes with retinoid X receptor (RXR) and translocates to the nucleus where the heterodimeric complex binds to vitamin D response elements (VDREs) to regulate the transcription of target genes [42,43,44,45,46]. Recently, retinoic acid-related orphan receptors (RORs) α and γ were shown to function as receptors for several vitamin D hydroxyderivatives [47,48]. RORs are involved in diverse cellular functions, including metabolism, immunity, and differentiation development [49,50,51]. Previously, we reported that RORα and RORγ are expressed in normal human skin and melanomas and that their expression decreases during the progression of melanocytic tumors [48,52]. The expression of these receptors correlated with intramural expression of HIF-1α (hypoxia-inducible factor 1 alpha) [53]. In addition, the reduced expression of RORγ was related to poorer melanoma prognosis [52].

In this study, we tested whether VDR, RORγ, and RORα are expressed in ovarian cancers, and we correlated their expression with clinico-pathological data, including patient survival time. Moreover, we investigated the effects of hydroxyderivatives of vitamin D and lumisterol, as well as RORγ/α agonists, on ovarian cancer cells cultured in vitro.

## 2. Materials and Methods

### 2.1. Clinico-Pathological Studies

#### 2.1.1. Patients

The ovarian cancer patients of the Oncology Center in Bydgoszcz, Poland treated between December 2006 and January 2010 were chosen consecutively from a patient digital database on the basis of pathomorphological diagnosis. After evaluation of quality and availability of histological material, 103 tissue blocks (including 65 ovarian tumors, 16 metastases, and 22 normal ovaries) from 68 patients were included in this study. The study was approved by the Institutional Review Board of Collegium Medicum, Nicolaus Copernicus University (KB446/2009 and KB344/2014) and adhered to the tenets of the Declaration of Helsinki. The clinico-pathomorphological characteristics of the patients included in this study are presented in Table 1.

#### 2.1.2. Immunohistochemistry

The expression of RORγ, VDR and RORα in ovarian tissues was analyzed by immune histochemistry, as described previously [52,54,55,56]. Briefly, after antigen retrieval, formalin-fixed, paraffin-embedded 4 μm sections were incubated with primary antibodies anti-RORγ (generated as previously described [48,57]), anti-VDR (clone 9A7, Millipore, Merck KGaA, Darmstadt, Germany), or anti-RORα (generated as previously described [48,57]) overnight at 4 °C. For the detection of RORγ and α, after extensive washing, the sections were incubated with secondary antibody EnVision™ FLEX/HRP (Dako, Carpinteria, CA, USA) and visualized with Vector NovaRED (Vector Laboratories Inc., Burlingame, CA, USA), before being counterstained with hematoxylin, dehydrated, and mounted in permanent medium (Consul Mount; Thermo Fisher Scientific Inc. Waltham, MA, USA). For the detection of VDR, after extensive washing, slides were incubated with anti-rat secondary antibody conjugated to HRP (Jackson ImmunoResearch Europe Ltd., Ely, UK), then visualized with ImmPACT NovaRED (Vector Laboratories Inc., Burlingame, CA, USA), and mounted in permanent medium (Consul Mount; Thermo Fisher Scientific Inc. Waltham, MA, USA). For each run of immunohistochemistry, negative and positive controls were prepared. In negative controls, primary antibody was omitted. Kidney samples served as positive controls for RORγ and VDR, and skin samples served as positive controls for RORα. Slides were analyzed using a Nikon Eclipse 80i light microscope (equipped with Nikon Digital Sight DS Fi1-U2 digital camera and NIS-Elements BR 3.0 software (Nikon Instruments Europe BV, Badhoevedorp, the Netherlands). The expression of CYP27B1 and RCAS1 (receptor binding cancer antigen expressed on SiSo cells) was detected using immunohistochemistry, as previously described [56,58,59].

#### 2.1.3. Immunohistochemistry Assessments

Clinical data, histopathological diagnosis, and other relevant data were blinded during immunohistochemical evaluation of immunostained sections. The immunostaining was analyzed semiquantitatively, with both percentage and immunostaining intensity evaluation. The semiquantitative score (SQ-score) was calculated as follows: SQ = mean (IR × SI), where IR is the percentage of immunopositive cells and SI is the staining intensity (assessed using the scale from 0 to 3 arbitrary units (AU). Since mean nuclear RORγ immunostaining was stronger than cytoplasmic staining, we applied different cut-off SQ-scores for those compartments. According to cytoplasmic RORγ SQ-score, patients were stratified as follows: SQ 0.0–50.0 = RORγ absent, SQ 50.1–75.00 = low RORγ, and SQ 75.1–300.0 = high RORγ. According to nuclear RORγ SQ-score patients were stratified as follows: SQ 0.0–75.0 = RORγ absent, SQ 75.1–100.0 = low RORγ, and SQ 100.1–300.0 = high RORγ. VDR-stained ovarian sections were stratified according to SQ-score as follows: SQ 0.0–69.99 = VDR absent, SQ 70.00–300.00 = VDR present for nuclear VDR, and SQ 0.0–50.00 = VDR absent, SQ 50.1–300.00 = VDR present for cytoplasmic VDR. Immunohistochemical evaluations of CYP27B1 and RCAS1 expression were performed using immunohistochemistry, as previously described [58,60]. Since the availability of the tumor tissue in the many OCs blocks was exhausted, clinico-pathologic correlation for RORα was not performed, and only descriptive data were presented.

### 2.2. Cell Culture-Based Studies

#### 2.2.1. Cell Culture and Treatment

The human ovarian epithelial adenocarcinoma cell lines SKOV-3 and OVCAR-3 were from the American Type Culture Collection (ATCC). The SKOV-3 cell line was cultured in Roswell Park Memorial Institute (RPMI) medium 1640 containing l-glutamine (Cellgro, Manassas, VA, USA). The OVCAR-3 cell line was maintained in Dulbecco’s modified Eagle’s medium (DMEM) medium (Cellgro, Manassas, VA, USA). Both media were supplemented with 10% fetal bovine serum (FBS; Atlanta Biologicals, Inc, Flowery Branch, GA, USA) and 1% antibiotic–antimycotic solution (Mediatech, Inc. Manassas, VA, USA). All cells were cultured in a humidified atmosphere with 5% CO_2_ at 37 °C.

Cells were treated with 1,25-dihydroxyvitamin-D_3_ (1,25(OH)_2_D_3_) (Sigma-Aldrich, St. Louis, MO, USA), 20-hydroxyvitamin-D3 (20(OH)D_3_), 20-hydroxy-lumisterol 3 (20(OH)L_3_), and the synthetic RORα/RORγ agonist, SR1078 (Sigma-Aldrich, St. Louis, MO, USA). 20(OH)D_3_ was enzymatically synthesized with a reconstituted CYP11A1-mediated system using vitamin D3 from Sigma-Aldrich (St. Louis, MO, USA) as the substrate [61,62]. 20(OH)L_3_ was a product of photochemical transformation of 20-hydroxy-7-dehydrocholestrol as described previously [63]. The products were purified by thin-layer chromatography, followed by reversed-phase high-performance liquid chromatography. Their molecular identities were confirmed by mass and UV spectra and NMR as described previously [62,63], and they were stored until use at −80 °C. All compounds being tested were dissolved in 100% ethanol. Control cells were treated with 0.1% ethanol, corresponding to the concentration of the ethanol in the treatment with 10^−6^ M 1,25(OH)_2_D_3_ or SR1078, or 10^−7^ M 20(OH)D_3_ or 20(OH)L_3_. During treatment, 5% charcoal/dextran-treated bovine serum (Atlanta Biologicals, Inc, Flowery Branch, GA, USA) was used.

#### 2.2.2. Fluorescent Immunocytochemistry (F-ICC)

Cells were seeded in a 96-well plate and grown until reaching ~70% confluence. Then, cells were fixed with 4% buffered formalin for 20 min at room temperature (RT), followed by permeabilization and blocking with 0.5% Triton X-100 and 2% bovine serum albumin (BSA) in phosphate-buffered saline (PBS) for 1 h at RT. Primary rabbit anti-RORγ antibody (developed at National Institute of Environmental Health Sciences (NIEHS, USA) [48,57]) diluted in 1% BSA solution was applied overnight at 4 °C. After extensive washing in PBS, the secondary antibody (goat anti-rabbit immunoglobulin G (IgG) (H + L) Cross-Adsorbed Secondary Antibody, Alexa Fluor 488, A-11008, Thermo Fisher Scientific, Inc., Waltham, MA, USA), diluted 1:1000 in 1% BSA solution was applied for 1 h at RT. Then, nuclei of cells were stained with propidium iodide (PI) (Sigma, Saint Louis, MO, USA) and examined with a Cytation 5 microplate reader, equipped with appropriate filters capable of visualization of green and blue fluorophores and Gene5 software (BioTek, Winooski, VT, USA). The positive control consisted of HaCaT cells. In the negative control, the primary antibody was omitted.

#### 2.2.3. Western Blot (WB)

OVCAR-3 and SKOV-3 cells were grown as described above, and protein extraction was performed as described previously [64]. Briefly, the cell extract was obtained by dissolving cell pellets in radioimmunoprecipitation assay (RIPA) buffer and incubating on ice for 10 min followed by centrifugation at 15,000× *g* for 10 min. The nuclear and cytoplasmic fractions were isolated separately with a Nuclear Extract Kit (Active Motif, Inc., Carlsbad, CA, USA) following the manufacturer’s protocol. Briefly, cells were suspended in hypotonic buffer and vortexed after adding detergent followed by centrifugation at 14,000× *g* for 30 s. The supernatant was used as the cytoplasmic fraction, and the pellet was resuspended in lysis buffer followed by incubating on ice for 30 min. The samples were centrifuged for 10 min at 14,000× *g*, and the supernatant was used as the nuclear fraction. Protein concentration was determined using a bicinchoninic acid (BCA) Protein Assay kit (Thermo Fisher Scientific, Waltham, MA, USA). Fifty micrograms of extracted protein was subjected to SDS/PAGE using a 4–15% acrylamide gel (Bio-Rad Laboratories, Inc., Hercules, CA, USA). Western blots were performed as described previously [41,52,64]. Briefly, after transfer of the proteins to a polyvinylidene fluoride (PVDF) membrane (Millipore Sigma, Burlington, MA, USA), the membrane was blocked with 5% nonfat dry milk followed by incubation with primary antibodies at a dilution of 1:200 for anti-VDR monoclonal antibody (D-6) (Santa Cruz Biotechnology, Dallas, TX, USA), 1:1000 for anti-RORα polyclonal antibody (Invitrogen, Waltham, MA, USA), or 1:500 for anti-RORγ (t) monoclonal antibody (Invitrogen, Waltham, MA, USA). As secondary antibodies, 1:4000 diluted anti-mouse IgG kappa binding protein conjugated to horseradish peroxidase (HRP) (Santa Cruz Biotechnology, Dallas, TX, USA) was used for VDR, while 1:5000 diluted goat anti-rabbit IgG (Abcam, Cambridge, MA, USA) or 1:2000 diluted anti-rat IgG HRP-linked antibody (Cell signaling Technology, Danvers, MA, USA) was used for RORα or RORγ, respectively.

#### 2.2.4. Cell Proliferation Assay

Cells were seeded in 96-well plates (TPP, Trasadingen, Switzerland) and grown until reaching ~70% confluence. Then, OVCAR-3 cells were serum-starved for 24 h, followed by treatment with the abovementioned compounds for 24 or 48 h. The starvation step was omitted in experiments using the SKOV-3 cell line. To assess the number of viable cells, the CellTiter 96 AQueous One Solution Cell Proliferation Assay (Promega Corporation, Madison, WI, USA) was used, according to the manufacturer’s protocol. Briefly, 20 µL of MTS (3-(4,5-dimethylthiazol-2-yl)-5-(3-carboxymethoxyphenyl)-2-(4-sulfophenyl)-2*H*-tetrazolium, inner salt) was added to each well, and plates were incubated for 2 h (SKOV-3) or 3 h (OVCAR-3) at 37 °C, followed by reading the absorbance at 490 nm using a Cytation 5 reader and Gene5 software (BioTek, Winooski, VT, USA). All experiments were performed in triplicate.

#### 2.2.5. Spheroid Formation

The spheroid assay was performed according to the protocol of Johnson et al. [65]. Briefly, SKOV-3 cells (400 cells per well) were seeded in RPMI-1640 medium (Cellgro, Manassas, VA, USA), supplemented with 20 ng/mL epidermal growth factor (Sigma-Aldrich), 10 ng/mL basic fibroblast growth factor (Sigma-Aldrich), 5 μg/mL insulin (Life Technologies, Invitrogen), 0.4% bovine serum albumin (Sigma-Aldrich), 1× B27 supplement (Life Technologies, Invitrogen), and the compounds to be tested, in ultra-low attachment 96-well plates and incubated for 7 days; subsequently, the plates were scanned using a Cytation 5 reader (BioTek, Winooski, VT, USA), and spheroids were counted using Gene5 software (BioTek, Winooski, VT, USA). All experiments were performed in triplicate. OVCAR-3 cells did not form spheroids using the abovementioned protocol.

### 2.3. Statistical Analysis

Prism 5.00 software (GraphPad Software, San Diego, CA, USA) was used for statistical analysis. One-way ANOVA or the *t*-test was used to compare two or more variables. The association between immunostaining and categorical variables was tested using Pearson’s correlation. The Kaplan–Meier method was used to analyze the survival data. Results were considered as statistically significant at *p* < 0.05.

## 3. Results

### 3.1. RORγ Expression in Human Ovarian Tumors Decreases

RORγ immunostaining was detected predominantly in the nuclei with weaker staining in the cytoplasm (Figure 1A–G. RORγ was found in both normal ovarian epithelial cells and primary and metastatic ovarian tumors; however, in pathological samples, its expression was significantly reduced (Figure 1A,B). For the ovarian cancer patient population included in this study, we analyzed 12 cases of matched pairs of primary and metastatic cancers, but no statistically significant differences were found. RORγ immunostaining decreased with decreasing differentiation levels of tumors. In primary tumors classified according to the cancer grade, there were statistically significant differences in nuclear RORγ levels observed among G1, G2, and G3 tumors (data not shown). Furthermore, those relationships were strongest in cancers classified according tumor grade in analyzed section. A gradual decrease was observed in both cytoplasmic and nuclear RORγ with increasing tumor grade (Figure 1C–F) indicating a strong negative correlation between the tumor grade and receptor expression (*r* = −0.35, *p* = 0.0024 and *r* = −0.024, *p* = 0.03, respectively). In metastatic tumors, we did not observe any relationship between RORγ immunostaining and tumor grade (Figure 1E,F).

Qualitative analysis could only be performed for RORα because of a limited availability of tumor tissue in paraffin blocks from many patients (Appendix A). RORα expression, similar to RORγ, was found predominantly in the nuclei with weaker staining in the cytoplasm (Figure 1G. RORα staining in ovarian cancers was heterogeneous and ranged from not detectable to strong nuclear and moderate cytoplasmic immunostaining.

### 3.2. RORγ Expression Correlates with CYP27B1 and RCAS1 Expression

CYP27B1 hydroxylates 25-hydroxyvitamin D_3_ at position C1α producing biologically active 1,25(OH)_2_D_3_ [32,33]. Since RORγ can act as receptor for vitamin D, we tested the relationship between CYP27B1 and RORγ expression. A high level of nuclear RORγ immunostaining (>100 AU) correlated with high CYP27B1 expression (Figure 2A). No significant relationship was observed between cytoplasmic RORγ and CYP27B1; CYP27B1 expression was the highest in cases with moderate cytoplasmic RORγ immunostaining (50–75 AU) (Figure 2B).

Estrogen receptor-binding site-associated antigen (EBAG9, also referred to as RCAS1) is a tumor-associated antigen that is expressed at high frequency in a variety of cancers. It promotes tumor progression by enhancing apoptosis in cytotoxic T lymphocytes, thereby facilitating immune escape and tumor progression [66,67,68]. Next, we analyzed the relationship between RORγ and RCAS1 expression and found an inverse correlation between their expression levels (*r* = −0.2688, *p* = 0.0152 and *r* = −0.3113, *p* = 0.0058 for cytoplasmic and membrane RCAS1, respectively) with significantly higher RCAS1 expression in cases with cytoplasmic RORγ levels of ≤50 AU (Figure 2C).

### 3.3. RORγ Expression Affects Survival of Ovarian Cancer Patients

RORγ expression is related to the tumor aggressiveness, and higher proliferation activity, as assessed by the percentage of Ki-67-positive cells, was found in cases with lower nuclear RORγ level (Appendix A). Analysis of the overall survival time (OS) revealed that higher RORγ expression in primary cancers was associated with longer overall survival (mean survival of patients without and with cytoplasmic RORγ was 33.4 vs. 43.9 months, respectively, while corresponding values for nuclear RORγ were 30.1 vs. 45.7 months, respectively). Furthermore, disease-free survival (DFS) was affected by disturbances in RORγ expression, and the mean DFS of patients without and with cytoplasmic RORγ was 0.0 vs. 18.8 months, respectively, while these values for nuclear RORγ were 2.4 vs. 16.3 months, respectively. In addition, Kaplan–Meier survival curves showed significant differences for both OS (Figure 3A,B) and DFS (Figure 3C,D) in relation to cytoplasmic (Figure 3A,C) and nuclear RORγ (Figure 3B,D). We did not observe significant differences in OS and DFS with respect to RORγ expression in metastatic cancers.

### 3.4. VDR Expression Decreases in Human Ovarian Tumors

Similar to RORγ, VDR immunostaining was found predominantly in cell nuclei with very limited cytoplasmic staining. In both primary and metastatic cancers VDR nuclear and cytoplasmic immunostaining decreased in comparison to normal epithelia (Figure 4A–G). In primary ovarian cancers, a decrease in nuclear VDR immunostaining was negatively correlated with the level of differentiation (*r* = −0.40, *p* = 0.0030), and the lowest VDR expression was observed in G3 tumors (Figure 4C,D). No correlation was found in nuclear and cytoplasmic VDR staining between primary lesions and metastatic tumors. Nuclear VDR immunostaining correlated negatively with expression of the proliferation marker Ki-67, but this correlation was limited to the central part of the tumor (*r* = −0.28, *p* = 0.035), defined as the older part of the tumor, with no signs of dynamic growth [60].

### 3.5. VDR Expression Affects Survival of Ovarian Cancer Patients

Analysis of the OS revealed that VDR expression in primary cancers was associated with longer overall survival (mean survival of patients without and with nuclear VDR was 33.6 vs. 48.8 months, respectively). The significance of these differences was supported by Kaplan–Meier survival curves (Figure 5). For cytoplasmic VDR, there were no differences in OS (not shown). In addition, DFS was not affected by VDR expression in ovarian cancer cells (not shown).

### 3.6. Expression of RORs and VDR in Human Ovarian Cancer Cell Lines

Both WB and immunofluorescence (IF) analyses showed expression of VDR, RORγ, and RORα in OVCAR-3 and SKOV-3 ovarian cancer cells, with the latter showing significantly stronger levels of expression (Figure 6A–F). Analysis of nuclear and cytoplasmic fractions probed with anti-VDR antibodies again showed a significantly higher signal in the nuclei from SKOV-3 cells (Figure 6A middle panel). Analysis of the cytoplasmic fraction from OVCAR-3 cells showed several bands with molecular weight (MW) higher than expected for VDR (Figure 6A right panel), suggesting processing through ubiquitination, which can give higher-MW species before final degradation [69].

### 3.7. Effect of Vitamin D3 and Lumisterol Hydroxyderivatives and of Synthetic RORα/γ Agonist, SR1078, on Growth of Ovarian Cancer Lines

Of the compounds tested, 20(OH)D_3_, 20(OH)L_3_, and SR1078, but not 1,25(OH)_2_D_3_, inhibited SKOV-3 growth in a dose-dependent manner as assessed by the MTS assay (Figure 7).

In OVCAR-3 cells 1,25(OH)_2_D_3_, 20(OH)L_3_, and SR1078 inhibited cell proliferation in a dose-dependent manner, whereas 20(OH)D_3_ had no significant effect (Figure 8).

The ability of cancer cells to form spheroids in vitro is often used as the assay to assess the anchorage-independent growth and/or stem-cell-like phenotype or the antitumor effect of drugs. Accordingly, we assessed the effects of 1,25(OH)_2_D_3_, 20(OH)D_3_, 20(OH)L_3_, and SR1078 on spheroid formation by the SKOV-3 cell line. Tests on OVCAR-3 cells were not performed since these cells did not form spheroids in vitro. While 1,25(OH)_2_D_3,_ 20(OH)L_3_, and SR1078 reduced spheroid formation in a dose-dependent manner, 20(OH)D_3_ showed no effect in the dose range 10^−9^ to 10^−7^ M (Figure 9).

## 4. Discussion

In this study, we analyzed the expression of RORα, RORγ, and VDR in ovarian tumor cells and identified an inverse correlation between their expression and a more aggressive phenotype and unfavorable clinical outcome. To the best of our knowledge, this is the first report showing the expression of RORγ in clinical OC samples. While the expression of VDR in OCs has been reported in several studies, some of these reports present contradictory data. Villena-Heinsen et al. [70] observed VDR expression in more than 83% of normal ovaries (*n* = 14) and 100% of ovarian cancers (*n* = 40). Similarly, Friedrich et al. [71] found an increased expression of VDR (assessed by immunohistochemistry and PCR) in cancers in comparison to normal ovaries. In our cancer samples and consistent with the observations by Thill et al. [72] and Cordes et al. [73], VDR expression was heterogeneous, but the semiquantitative analysis indicated lower VDR levels in cancers than in normal ovaries (Figure 4). While the paper was under review Pejovic et al. [74] published similar association on vitamin D signaling and ovarian cancer. Reduced VDR expression in malignant tissues was also found in other tumors, such as cutaneous and uveal melanomas [54,55,75], urothelial bladder [56], colon [76], and lung [77] cancers. In some tumors, the reduced expression of VDR correlated to worse prognosis. 

Similarly to VDR, we also showed lower RORγ expression in ovarian cancers in comparison to normal tissues. This decreased RORγ level was associated with a poorer prognosis (Figure 1 and Figure 3). These observations are in agreement with previous reports showing the reduced expression of RORs in other tumors, such as cutaneous and uveal melanomas [48,52,75], hepatoma cells [78], and breast cancers [79]. Increased RORγ expression corresponded to a better prognosis and clinical outcome for patients with the above tumors [52,75,80]. Our previous study showed reduced CYP27B1 levels in primary and metastatic ovarian cancers compared normal controls, with the strongest reduction found in cancers showing a high proliferation index, lack of immune response, presence of necrosis, and dynamic tumor growth. In addition, a lack of CYP27B1 expression affected the overall survival time [40]. The current study uncovered a correlation between higher RORγ expression and higher CYP27B1 expression (Figure 2).

We also assessed the in vitro phenotypic impact of 1,25(OH)_2_D_3_, 20(OH)D_3_, 20(OH)L_3_, and SR1078 on OC cells and found that the inhibition of proliferation was affected by the chemical nature of the molecules tested (Figure 7, Figure 8 and Figure 9). This anticancer activity of vitamin D derivatives against OC cells is in line with other studies showing a reduction in cancer cell proliferation, but some differences in these effects were observed. 20(OH)L_3_ and synthetic RORα/RORγ agonist, SR1078, inhibited proliferation in both cell lines, while 20(OH)D_3_ only inhibited the SKOV-3 cell proliferation. 1,25(OH)_2_D_3_, 20(OH)L_3_, and SR1078, but not 20(OH)D_3_, inhibited spheroid formation in SKOV-3 cells (Figure 7, Figure 8 and Figure 9). The variation in the response of OVCAR-3 and SKOV-3 cell lines could result from the differential expression of VDR, RORα, and RORγ (Figure 6). Previously, we showed that 1,20(OH)_2_D_3_ and 20(OH)D_3_ inhibit the proliferation of human melanomas cells and CAL-27 oral squamous cell carcinoma [20,21,81,82]. The antitumor activity of SR1078 was previously reported for MCF-7 and MDA-MB-231 breast cancer cells [79]. Our results for vitamin D hydroxyderivatives are in agreement with previous publications regarding the effects of vitamin D and its derivatives on ovarian cancer cells. In mouse ovarian surface epithelial cells, calcitriol delayed malignant transformation induced with 7,12-dimethylbenz[a]anthracene (DMBA) and decreased colony formation in soft agar [83]. Vitamin D stimulated the expression of VDR and reduced the metastatic potential of ovarian cancer cells by increasing E-cadherin and decreasing the expression of β-catenin both in vivo and in vitro [83], while, in OVCAR-3 and SKOV-3 ovarian cancer cells, calcitriol showed antiproliferative effects with an additive effect with celecoxib [84]. Furthermore, the growth of OVCAR-3 tumor xenografts in nude mice was inhibited by the vitamin D analogue, EB1089 (seocalcitol), while calcitriol and EB1089 inhibited the growth of OVCAR-3 cells [85]. OVCAR-3 cells are also sensitive to inhibition of migration and invasion into the omentum by calcitriol in ex vivo mouse models [86]. The expression of VDR in cancer and in stromal cells is important for the inhibition of invasion of OC cells by vitamin D or its analogue, EB1089 (as found in VDR null mouse) [86]. The inhibition of cancer growth by calcitriol can be mediated via regulation of miR-498 and telomerase activity [87]. Correspondingly, the ROR agonist, SR1078, was able to enhance the anticancer effect of vitamin D [88]. In addition, SR1078 stabilized p53 in cancer cells, thereby increasing p53 function and apoptosis [89].

## 5. Conclusions

In conclusion, our results demonstrate the anticancer properties of 1,25(OH)_2_D_3_, 20(OH)D_3_, and20(OH)L_3_ against OCs cells. Importantly, the decreased expression of RORγ and VDR in ovarian tumor cells correlates with a more aggressive behavior and negative patient outcome. We also suggest that modulation of VDR and RORγ activities with their agonists could affect ovarian cancer cell behavior, suggesting their usefulness as potential targets in ovarian cancer therapy.

## Figures and Tables

**Figure 1 nutrients-12-03541-f001:**
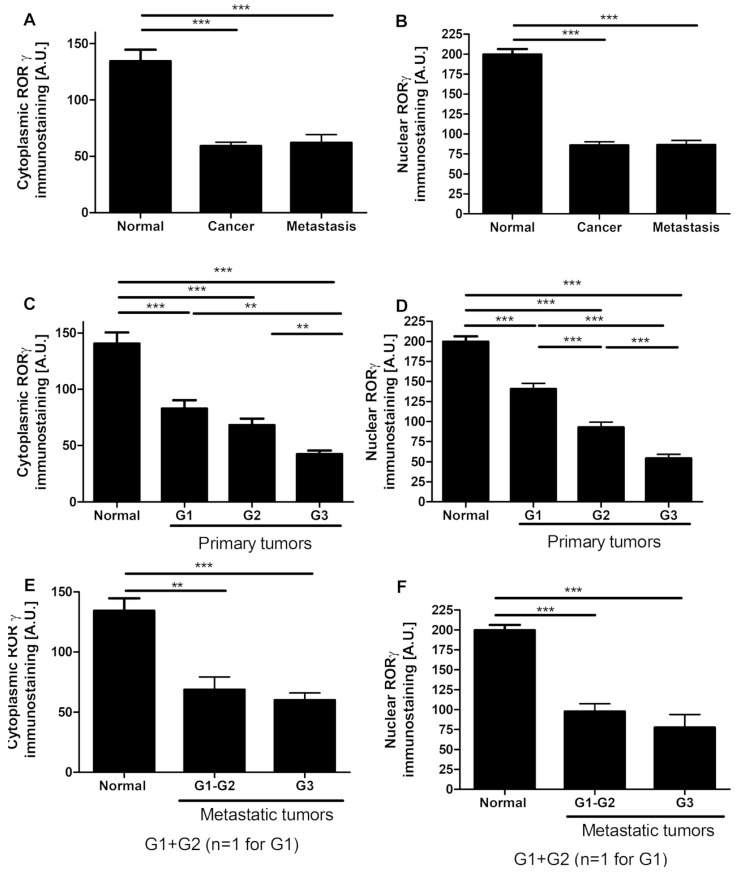
Cytoplasmic and nuclear retinoic acid-related orphan receptor γ and α (RORγ and RORα) immunostaining in normal ovary (*n* = 22) and ovarian tumors (*n* = 65) (**A**,**B**), in relation to tumor grade in primary tumors (**C**,**D**) and in metastatic tumors (*n* = 16) (**E**,**F**). Statistically significant differences are denoted with asterisks as determined by ANOVA (** *p* < 0.01, and *** *p* < 0.001). (**G**) Representative RORγ immunostaining in normal epithelia, ovarian cancers, and positive (POS CTRL) and negative (NEG CTRL) controls. Scale bars = 50 µm.

**Figure 2 nutrients-12-03541-f002:**
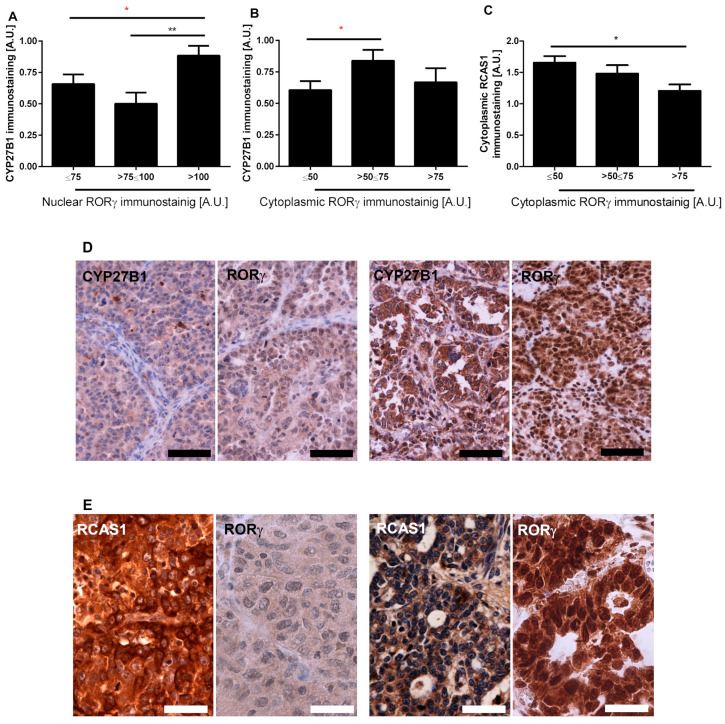
Correlation of nuclear (**A**) and cytoplasmic (**B**,**C**) RORγ immunostaining to cytochrome P450 (CYP)27B1 (**A**,**B**) and RCAS1 (receptor binding cancer antigen expressed on SiSo cells) (**C**) expression in ovarian tumors (*n* = 63). Statistically significant differences are denoted with asterisks as determined by ANOVA (black asterisks) or Student’s *t*-test (red asterisks, * *p* < 0.05, ** *p* < 0.01). (**D**) Representative images of CYP27B1 and RORγ immunostaining of two ovarian cancer cases. (**E**) Representative images of RCAS1 and RORγ immunostaining of two ovarian cancer cases. Scale bars: black = 50 µm, white = 100 µm.

**Figure 3 nutrients-12-03541-f003:**
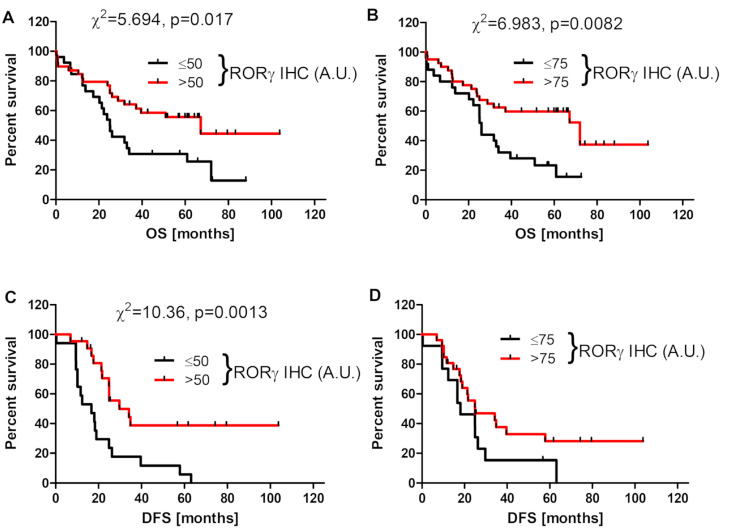
Overall survival (OS) (**A**,**B**) and disease-free survival (DFS) (**C**,**D**) of ovarian cancer patients in relation to cytoplasmic (**A**,**C**) and nuclear (**B**,**D**) RORγ immunostaining (*n* = 65).

**Figure 4 nutrients-12-03541-f004:**
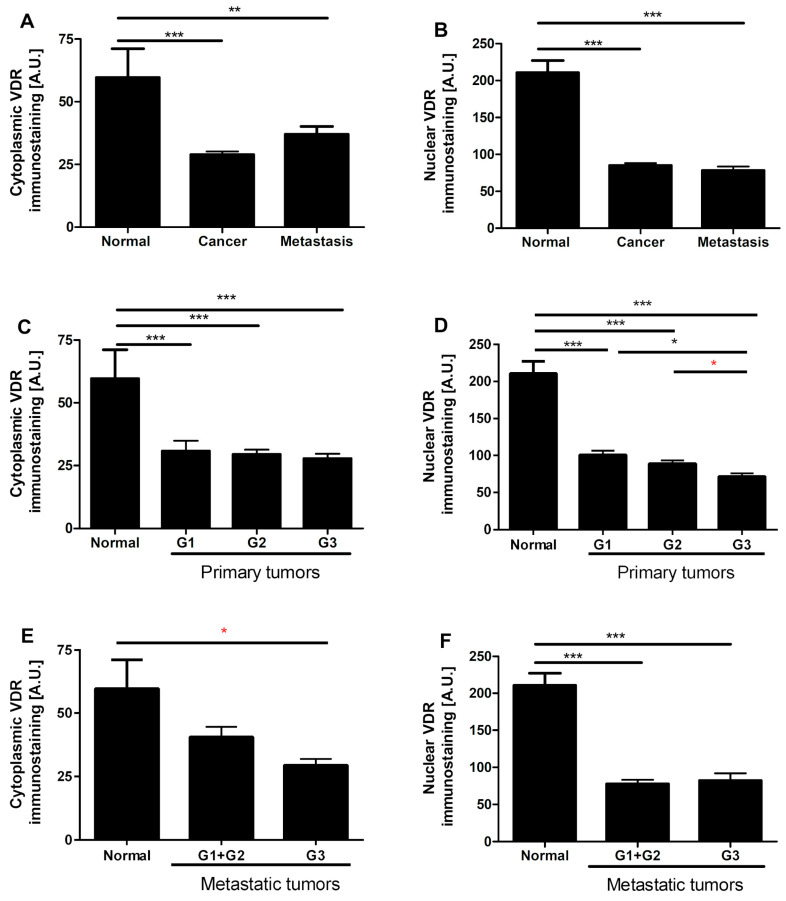
Cytoplasmic and nuclear vitamin D receptor (VDR) immunostaining in normal ovary and ovarian tumors (*n* = 20) (**A**,**B**), in relation to tumors grade in primary tumors (*n* = 57) (**C**,**D**) and in metastatic tumors (*n* = 13) (**E**,**F**). Statistically significant differences are denoted with asterisks as determined by ANOVA (black asterisks) or Student’s *t*-test (red asterisks, * *p* < 0.05, ** *p* < 0.01, *** *p* < 0.001). (**G**) Representative VDR immunostaining in normal epithelia, ovarian cancers, and positive (POS CTRL) and negative (NEG CTRL) controls. Scale bars = 50 µm.

**Figure 5 nutrients-12-03541-f005:**
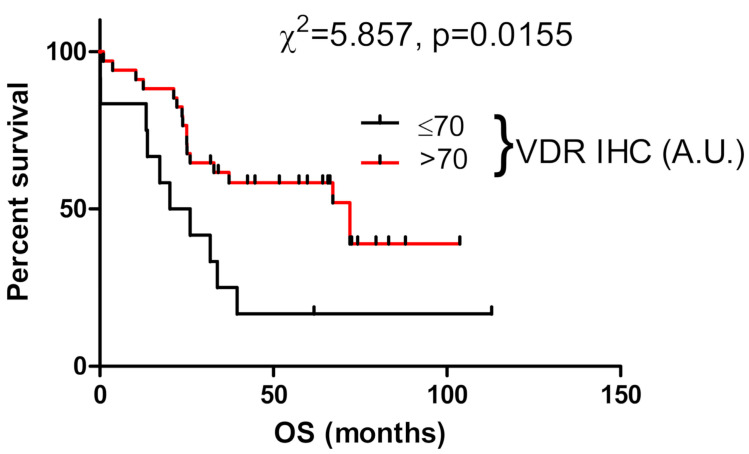
Overall survival (OS) of ovarian cancer patients in relation to nuclear VDR immunostaining (*n* = 57).

**Figure 6 nutrients-12-03541-f006:**
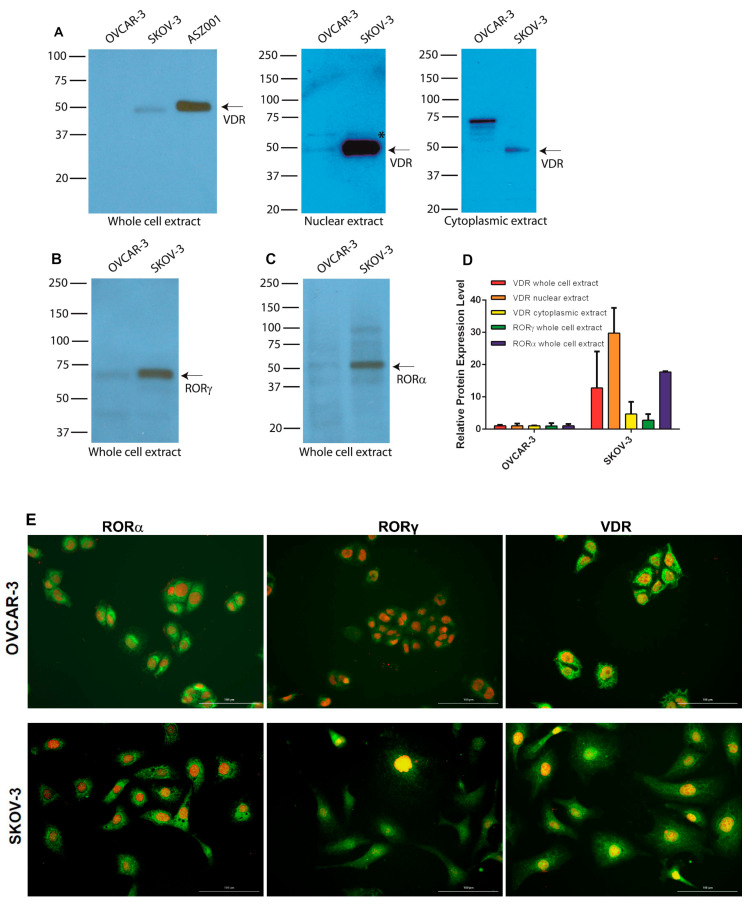
RORα, RORγ, and VDR in OVCAR-3 and SKOV-3 ovarian cancer cells. Expression of VDR and RORα/γ in ovarian cancer cell lines as assessed by Western blot (WB): (**A**) VDR; (**B**) RORγ; (**C**) RORα. (**D**) Relative protein expression in OVCAR-3 and SKOV-3 as measured by WB in whole-cell extracts (*n* = 2) and nuclear (*n* = 2) and cytoplasmic (*n* = 2) fractions. ASZ001 (mouse basal carcinoma cell line) was used as a positive control for VDR expression in panel (**A**). (**E**) Immunofluorescence detection of RORα, RORγ, and VDR in OVCAR-3 and SKOV-3 ovarian cancer cells. Scale bars = 100 μm. * *p* < 0.05.

**Figure 7 nutrients-12-03541-f007:**
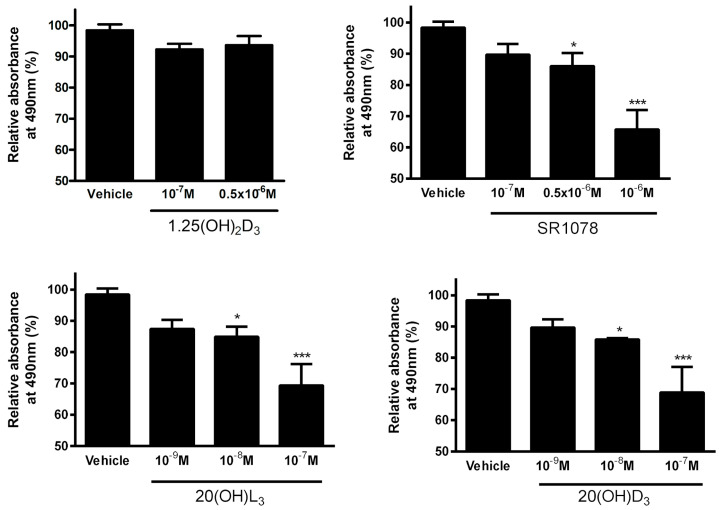
Effects of 1,25(OH)2D3, 20(OH)D3, 20(OH)L_3_, and SR1078 on proliferation of SKOV3 ovarian cancer cell line as assayed with the MTS (3-(4,5-dimethylthiazol-2-yl)-5-(3-carboxymethoxyphenyl)-2-(4-sulfophenyl)-2*H*-tetrazolium) test (*n* = 3). Data are presented as a percentage of the control. Statistically significant differences are denoted with asterisks as determined by ANOVA (* *p* < 0.05, *** *p* < 0.001).

**Figure 8 nutrients-12-03541-f008:**
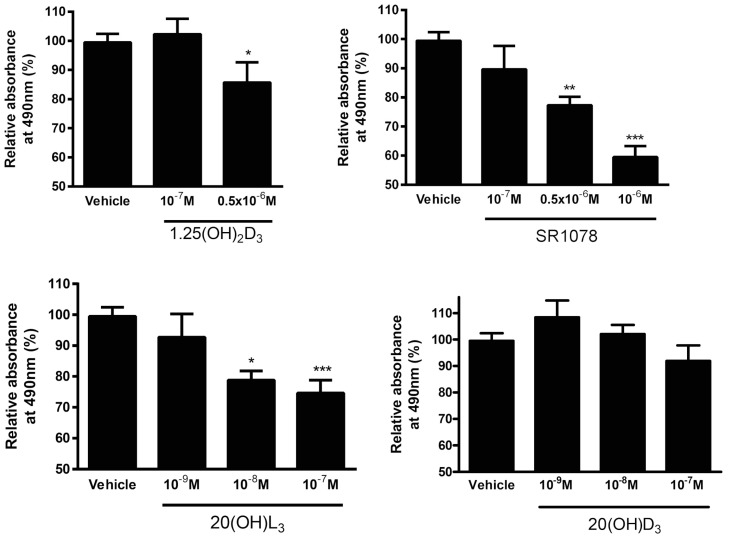
Effect of 1,25(OH)2D3, 20(OH)D3, 20(OH)L_3_, and SR1078 on proliferation of OVCAR3 ovarian cancer cell line as assayed with MTS test (*n* = 3). Data are presented as a percentage of the control. Statistically significant differences are denoted with asterisks as determined by ANOVA (* *p* < 0.05, ** *p* < 0.01, and *** *p* < 0.001).

**Figure 9 nutrients-12-03541-f009:**
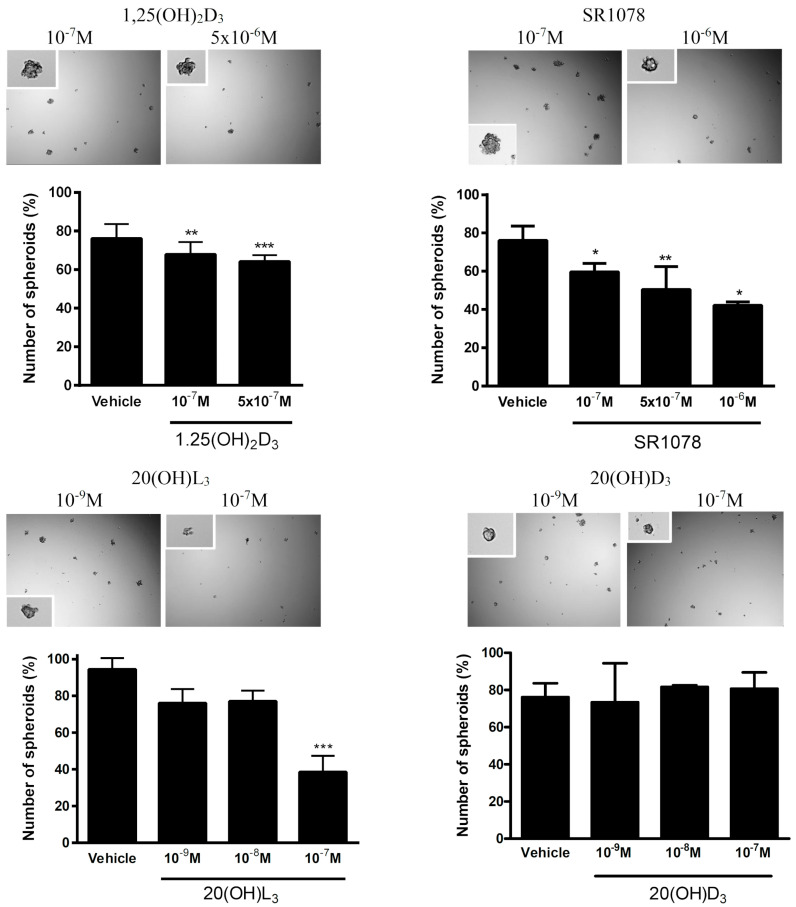
Effect of 1,25(OH)_2_D_3_, 20(OH)D_3_, 20(OH)L_3_, and SR1078 spheroid formation by SKOV-3 ovarian cancer cell line (*n* = 3). Data are presented as a percentage of the control. Statistically significant differences are denoted with asterisks as determined by ANOVA (* *p* < 0.05, ** *p* < 0.01, and *** *p* < 0.001). Images present representative spheroids, and insets present high-magnification images of single spheroids.

**Table 1 nutrients-12-03541-t001:** Pathomorphological characteristic of the patients included in the present study.

Feature	Number of Cases
Age	Mean 55.8 years (25.6–84.3)
<40 years	3
41–50 years	14
51–60 years	36
61–70 years	10
>70 years	5
Grading *	
G1	3
G2	19
G3	43
Grading **	
G1	0
G2	9
G3	7
Histological type *	
Borderline	2
Serous adenocarcinoma	50
Clear-cell carcinoma	3
Endometrioid cancer	6
Mucinous cancer	1
Transitional cell carcinoma	1
Other	2
Necrosis *	
Absent	27
Present	38
Metastases ^#^	
Absent	14
Present	51

* primary tumors; ** metastatic tumors; ^#^ metastases present at the time of cytoreductive surgery.

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
