# Peer review of "Association among Vitamin D, Retinoic Acid-Related Orphan Receptors, and Vitamin D Hydroxyderivatives in Ovarian Cancer"

_nutrients, 2020, doi:10.3390/nu12113541_

Round 1

Reviewer 1 Report

The manuscript submitted by Anna A. Brożyna and colleagues, “On the role of vitamin D and retinoic acid-related orphan receptors a and g in ovarian cancer, describes the location and production of vitamin D receptor and vitamin D hydroxyderivatives in human ovarian cancer and how the vitamin D hydroxyderivatives impact the cell growth of ovarian cancer cells. The authors conclude that these receptors can be relevant for the diagnosis and prognosis of ovarian cancer. The work has a potential interest and fits with the scope of the journal. However, some issues should be clarified particularly regarding in vitro results.

Major concerns

  1. As mentioned in the manuscript, the samples for the RORa were limited. However, the authors can provide the quantification of the processed samples as supplementary material.
  2. The authors suggested that vitamin D and retinoic acid-related orphan receptors decrease with tumor aggressiveness. Some correlation with KI67 should be provided, at least as supplementary material.
  3. Figure 2: Although the results with CYP27B1 were already published, representative figures should be provided in the same samples (supplementary material). In addition, a correlation with tumor grades (not only with levels of ROR) would improve the quality and interpretation of the results.
  4. The impact of the different treatments and the protein levels of the receptors on OVCAR-3 and SKOV-3 cells was not similar. The differences between both cell lines should be indicated in the Discussion and relate them with the results.
  5. Figure 6: It is necessary to include the quantification of the western-blots (3 experiments).
  6. The analysis of the protein levels of VDR and ROR receptors after the different treatments would improve the quality of the manuscript.
  7. For most of the VitC derivatives, three concentrations are used, except for 1.25(OH)2D3. What is the effect of higher concentration?
  8. The abstract needs to be improved, particularly the introduction. Some results can also be better introduced (especially section 3.1 and 3.4)

Minor concerns

  1. The number of samples for each experiment should be included in the legend figures.
  2. Line 99: It is necessary to mention the Helsinki declaration
  3. Figures 1 and 4: The grades should be indicated in the representative samples, avoiding the considerable amount of letters.
  4. Line 278: there is a double space.
  5. Figure 4: Unify criteria for the statistics (astheric or the p-value)
  6. Figure 6: Put the name of the cell instead of the numbers

Author Response

Thank you kindly for your time and consideration regarding our manuscript. The  point-by-point response to the feedback we received is in the attached file. All changes in the manuscript are made with Track Changes option and seen as red font.

Reviewer 2 Report

The author did a very good job to design this manuscript “On the role of vitamin D and retinoic acid-related orphan receptors α and γ in ovarian cancer” and it is looking sound, elaborately discussed, categorized very well in different section to cover the role of vitamin D receptor (VDR) and retinoic acid-related orphan receptors γ and α in ovarian cancer.

Minor correction:

  • The title needs to be correct, as it starts with “On the role….” which is not appropriate and not looking sound.
  • At line 401, there is a duplication of words “the anti-cancer properties of anti-cancer properties of” needs to be correct.

Author Response

(The authors gave the same response as above.)

Round 2

Reviewer 1 Report

In the new version submitted, the authors have included all the information and changes suggested. Thus, the quality of the work already presents the qualitative requeriments for its publication in Nutrients.